# Epistemic Trust, Mistrust and Credulity Questionnaire (ETMCQ) validation in French language: Exploring links to loneliness

Christian Greiner[1]*, Vincent Besch[1,2], Marissa Bouchard-Boivin[1,3],
Catherine Le Hénaff[1], Cécilia Von Rohr-De Pree[1], Nader Perroud[1,4], Paco Prada[1,4],
Martin Debbané[2,5]

1 Department of Psychiatry, Geneva University Hospital, Geneva, Switzerland, 2 Faculty of Psychology and Educational Science, Geneva University, Geneva, Switzerland, 3 Montreal Mental Health University Institute, Montreal, Canada, 4 Faculty of Medicine, Geneva University, Geneva, Switzerland, 5 Research Department of Clinical, Educational and Health Psychology, University College London, United Kingdom

☺ These authors contributed equally to this work.
* christian.greiner@hug.ch

## Abstract

The concept of epistemic trust is gaining traction in the mental health field. Epistemic trust is thought to play a foundational role as a resilience factor against the development and maintenance of psychopathology by fostering social learning. The primary aim of this study was to validate the French-language version of the Epistemic Trust, Mistrust and Credulity Questionnaire (ETMCQ). We further sought to replicate previously reported associations with key developmental and psychological factors (childhood trauma, mentalizing and attachment) and test for epistemic trust's potential mediating roles between childhood traumatic experiences and psychopathology, and between loneliness and psychopathology. A total of 302 participants were recruited via the online survey platform Prolific. Confirmatory factor analysis and generalized linear models of mediation were conducted. Our findings suggest that the ETMCQ is a valid instrument to assess epistemic trust in the French language. Satisfactory psychometric properties were found to replicate the original three-factor solution in a Francophone population with a 12-item version of the questionnaire, with criterion-related validity similar to that previously published in validations of the ETMCQ in other languages. We also replicate previous findings reporting differential associations between epistemic stances (trust, mistrust and credulity) and attachment dimensions and styles, while also replicating mediation analyses showing the role of epistemic stances in the relationship between childhood traumatic experiences and psychopathology. Finally, we report preliminary evidence suggesting that epistemic trust mediates the well-described association between loneliness and psychopathology. Future research should investigate the ETMCQ in clinical populations in which psychopathological expressions are severe, enduring and co-occurring, where identifying potential mediators could help target and personalize psychosocial interventions.

**Data availability statement:** All relevant data are within the paper and its Supporting Information files.

**Funding:** The author(s) received no specific funding for this work.

**Competing interests:** The authors have declared that no competing interests exist

## Introduction

In the past decade, the field of mental health has increasingly focused on social-cognitive abilities enabling individuals to form, maintain, lean on and learn from interpersonal relationships and social networks [1]. In this vein, Fonagy et al. [2] initially argued that within early attachment relationships, the ability to mentalize (reflective functioning) – that is, to perceive, coordinate and interpret the behaviors of the self and others as being based on mental states – plays a crucial role in fostering resilience. More recently, they and others proposed re-examining the developmental relationship between early adversity and the expression of psychopathology over the lifespan by examining the dynamics among attachment, mentalizing and trust [3–6]. Building on a broad range of disciplines such as evolutionary anthropology [7], philosophy [8,9] and epistemology [10], as well as developmental psychology [11] and attachment research [12,13], the authors have broadened their mentalization-based theoretical framework to propose an integrative model in which early adversity in attachment relationships, impaired mentalizing and psychopathology are developmentally interwoven through difficulties in attributing trustworthiness to key sources of social information, that is, to attribute *epistemic trust* [14]. They suggest that attachment adversity breeds patterns of mistrust and credulity, ultimately limiting social learning and consequently interfering with adaptation and resilience. Epistemic trust is thus shaped by early attachment experiences, with secure attachments more likely to foster the development of trust in the reliability of knowledge communicated by others. Insecure attachments, conversely, can lead to challenges in evaluating reliability, potentially resulting in either excessive doubt or an uncritical acceptance of information [14]. The mentalizing capacities of attachment figures serve as a signaling system (through ostensive communication [11]) or a "trigger to trust", which assists the child to identify trustworthy sources of social knowledge.

To empirically evaluate these hypotheses, Campbell et al. [15] designed a scale to measure epistemic trust. They define epistemic trust as "the capacity of the individual to consider knowledge that is conveyed by others as significant, relevant to the self, and generalizable to other contexts" ([15], p. 2). Building on this definition, Campbell et al. [15] operationalized the concept through three relevant epistemic stances embodied within social communication: epistemic trust (ET), epistemic mistrust (EM) and epistemic credulity (EC). ET is characterized by selective and balanced receptivity to social learning opportunities within relationships, and the capacity to maintain confidence in the reliability and value of information from others. EM is defined as perceiving a source of information as untrustworthy, favoring a tendency to remain impermeable to the influence of others in social communication. EC entails decreased vigilance and discrimination with regard to social signals, making the individual prone to misinformation and exploitation [16]. Campbell et al. [15] created the first self-report instrument, the Epistemic Trust, Mistrust and Credulity Questionnaire (ETMCQ), offering evidence for a three-factor structure (ET, EM and EC), empirically substantiating the hypothesis of three distinct epistemic stances, and yielding acceptable reliability and validity indices. They reported significant associations between both EM and EC and higher levels of childhood adversity, lower mentalizing capacities, and more frequent insecure attachment patterns and expressions of psychopathology. Their post-hoc analyses suggest differential associations between attachment styles and epistemic stances. Interestingly, in their study, both EM and EC partially mediated the link between early adversity and expression of psychopathology. This original study has been followed by at least four independent ETMCQ translation and replication studies. Liotti et al. [17] translated the scale into Italian and validated it, providing evidence for the instrument's cross-cultural applicability, by reporting a three-factor structure similar to the model proposed in the original validation, albeit with slight differences on item loadings and

total item number (14-items). The authors found that EM and EC but not ET were significantly related to the presence of childhood traumatic experiences, attachment avoidance and anxiety, lower levels of mentalizing, lower emotional-regulation capacities, and higher levels of psychopathological symptoms. Asgarizadeh et al. [18] adapted the scale into Persian and reported criterion-related validity for EM and EC but not for ET, and an acceptable internal consistency for EC but not for ET and EM. They found discriminant power for EM and EC in detecting positive screens for borderline personality disorder, and measurement invariance across sexes. Rodriguez Quiroga et al. [19] validated an Argentine version of the scale with a satisfactory three-factor solution with correlating residuals, with stable scores over time. Significant positive correlations were found with anxious and fearful-avoidant attachment, hypomentalizing (reduced attribution of mental states), childhood traumatic experiences, and expression of psychopathology. Weiland et al. [20] validated the scale in German with a shortened 12-item version. The study found links between EM and EC with markers of psychopathology but was unable to identify any significant differences between clinical and non-clinical participants. For EM and EC, correlations with associated constructs supported their construct validity, the results for ET were however heterogeneous. Other studies using the ETMCQ in diverse populations have provided evidence linking EM and EC scores to adverse childhood experiences, insecure attachment, poor mentalizing, and psychopathology [16,17,21–29].

Importantly, the recent conceptual developments of the mentalization-based model relate the association between trauma and epistemic trust to the experience of loneliness. In this context, loneliness may represent both a central subjective experience – "aloneness of mind" – and a social positioning – "isolation" – contributing to the development and maintenance of psychopathology [3]. Three pivotal arguments justify the incorporation of loneliness as a central theoretical pillar in the mentalization-based model. First, loneliness is thought to represent a paradigmatic mental state at the intersection of traumatic experiences and the disruption of social communication. Fonagy and colleagues [30] argue that traumatic events become particularly damaging when they are experienced in a state of subjective aloneness, when an individual feels that their mental experience cannot be shared, leaving the mind "alone" at the heart of the helplessness experienced in relation to trauma. In such circumstances, the essential process of calibrating one's mind through a mentalizing interaction with others is lacking. When available, social referencing through mentalizing is known to mitigate the impact of traumatic experiences by allowing individuals to reframe and contextualize overwhelming experience. By contrast, when such social referencing is absent, the process of reappraisal is crucially impaired, promoting a cycle of increased vulnerability and isolation [31]. From a mentalization-based standpoint, this process is accompanied by a reluctance or even a breakdown in the attribution of epistemic trust, preventing the essential mutual mentalizing needed to solicit and benefit from the support of others. This, in turn, reinforces the sense of subjective loneliness – that is, perpetuating a sense of being fundamentally alone in one's mental experience – which behaviorally reinforces isolation. Building on this first argument, Fonagy and colleagues reframe borderline personality disorder (BPD) through the lens of epistemic stances, characterizing the disorder as a condition related to social connectedness: "BPD describes a state of social inaccessibility. It can be conceived of as a temporary state of incompatibility with a grand evolutionary design of intra-cultural communication in which we all play a part. It describes a state of isolation from communication with one's partner, one's therapist, one's teacher, all created by epistemic mistrust." ([4], p. 592). Here, loneliness is conceived as a key barrier to social learning. The third argument proposes that shared intentionality with other social agents, the so-called "we-mode", is a key process sustaining the sense of belonging and agency in any individual's social involvement [32]. Loneliness in the face of trauma, which reinforces stances of mistrust and credulity, critically

impedes engagement in "we-mode" socialization. Developmentally, this can lead to deep-rooted feelings of not belonging and helplessness, which are known to increase vulnerability for psychopathology. Long before its incorporation into the theoretical framework of mentalizing, the concept of loneliness was explored extensively in research [33–38] documenting its role as a risk factor for myriad physical and mental health conditions, rivaling well-established morbidity risk factors such as physical inactivity, smoking and obesity [39–42]. In the current study, we sought to test the hypothesis that epistemic stances act as significant mediators in the relationship between loneliness and psychopathology.

## Aims and hypotheses

The overall aim of this study was to validate the French-language version of the ETMCQ. First, we hypothesized that we would reproduce the three-factor structure, composed of ET, EM and EC as reported in previous studies, as well as comparable results in terms of internal consistency and internal and external validity. Second, we aimed to replicate the findings of Campbell et al. [15] showing that the ETMCQ subscales differentially associate with distinct attachment dimensions and styles. Third, we sought to investigate the mediating role of the epistemic stances on the association between childhood trauma and psychopathology, similarly to Campbell et al.'s [15] analysis. Finally, we aimed to examine the mediating role of the epistemic stances in the relationship between loneliness and psychopathology.

## Methods

### Procedure

For this study, 372 participants were recruited via the online survey platform Prolific (https://www.prolific.com/). The required sample size was calculated on the basis of the results of Campbell et al. [15], who found the smallest factor loading to be 0.406, and on the recommendations of Jackson et al. [43] supporting that, in the case of a confirmatory factor analysis (CFA) with 3 factors and 5 items on each factor, a sample size of 200 is sufficient to obtain a satisfactory convergence. The inclusion criteria were that participants must be 18 years of age or older, fluent in French, and had never been hospitalized for psychiatric reasons. Participants received a financial compensation of 6.00 CHF for each test session, with the retest incremented by 25% at 1-year interval. To ensure the quality of the data the participants provided, several additional selection criteria were applied. First, participants needed to have satisfactorily submitted at least 15 online surveys beforehand to ensure a good level of use of the online platform and reduce data entry errors. Then, three bogus items were randomly inserted into the survey and response times were monitored in order to check participants' linguistic skills and attention [44]. Participants were informed of these attention and time controls before taking the survey. Participants were excluded if they gave incorrect answers to more than two bogus items, or if their response time was faster than the mean – 2 sd of the response times measured on the whole sample. Participants who gave an incorrect answer to one bogus item were excluded if their response time was faster than the mean – 1 sd. In total, 70 participants failed these attention control criteria and were excluded from the analyses. The final sample used for analyses was $N = 302$ participants. All participants provided sociodemographic data and completed the initial set of 15 items of the ETMCQ and additional validity questionnaires (see S2 and S3 Appendix for the complete dataset). The recruitment period started on 21 April 2022, and ended on 19 January 2024. The study was approved by the Swiss Ethics Commission in Geneva under project ID 2021-01100. Participants gave electronic consent for their involvement in the study and the use of their data for analysis and publication in a format approved by the Commission. The validation process of the French-language ETMCQ began with the

translation of the original English ETMCQ by independent individuals with French or English as their first language (S1 Appendix), following a forward-backward-forward procedure [45].

## Measures

The Epistemic Trust, Mistrust and Credulity Questionnaire (ETMCQ) [15] is a 15-items self-report scale to assess epistemic trust, using a Likert-scale ranging from 1 (strongly agree) to 7 (strongly disagree). The ETMCQ evaluates three core dimensions, ET which measures the readiness to accept information as reliable, EM reflecting skepticism towards information sources, and EC indicating a tendency to accept information uncritically.

The Childhood Trauma Questionnaire - Short Form (CTQ-SF) [46] is a 28-item self-report instrument validated for clinical and non-clinical populations. The questionnaire evaluates the presence and severity of five types of childhood traumatic experiences: physical abuse, emotional abuse, sexual abuse, emotional neglect and physical neglect. Individuals are asked to indicate on a 5-point Likert scale how often they suffered specific traumatic childhood experiences. The French validation of the instrument demonstrated that the five types of traumatic childhood experiences are valid and usable with French-speaking populations [47].

The Reflective Function Questionnaire (RFQ-8) [48] is an 8-item self-report instrument that evaluates mentalizing capacities by the degree of certainty and uncertainty with which individuals use mental-state information to understand their own and others' behavior. Subjects are asked to express their agreement with each item on a 7-point Likert scale. The uncertainty about mental states subscale (RFQ_U) captures poor use of mental-state information and a stance characterized by a lack of knowledge about mental states. The certainty about mental states subscale (RFQ_C) captures better use of mental-state information and adaptative levels of certainty about mental states. The French validation of the instrument demonstrated satisfactory reliability and construct validity of the two subscales [49].

The Experience in Close Relationships Scale-Revised (ECR-R) [50] is a 36-item questionnaire exploring adult attachment style. Items are rated on a 7-point Likert scale. The instrument comprises two subscales measuring two attachment dimensions: anxiety (intense concern for relationships, fear of abandonment) and avoidance (feelings of discomfort in establishing emotional closeness with a partner, difficulty in trusting others). Four attachment styles can be defined according to individuals' anxiety and avoidance scores [51,52]: a "Secure" attachment style is associated with low scores on both anxiety and avoidance, "Fearful" with high scores on both anxiety and avoidance, "Preoccupied" with high anxiety and low avoidance, and "Dismissing" with high avoidance and low anxiety. The French validation of the questionnaire has good reliability and validity and is consistent with its theoretical model of attachment dimensions [53,54].

The University of California Los Angeles Loneliness Scale (UCLA-LS) [55] is a 20-item scale designed to measure a participant's subjective feeling of loneliness and of social isolation on a 4-point Likert scale. The scale is conceptualized as unidimensional in its structure. The scale items tap into both the frequency and intensity of salient aspects and events of the experience of loneliness (e.g., "How often do you feel alone?"). Evaluation of the scale found high internal consistency and good validity [36]. It is the most commonly used self-report loneliness instrument: a meta-analysis looking at 149 studies found that the UCLA-LS was used in 27% of the studies, far more than for any other formal scale [36,41]. The French validation of the instrument has demonstrated good internal consistency and construct validity [56].

The Symptom Checklist-90-Revised (SCL-90-R) [57] is a 90-item instrument that evaluates a broad range of psychological problems and symptoms of psychopathology on a

5-point Likert scale. It yields nine scores along primary symptom dimensions (somatization, obsessive-compulsive, interpersonal sensitivity, depression, anxiety, hostility, phobic anxiety, paranoid ideation and psychoticism) and three scores on global distress indices (global severity index [GSI], hardiness and symptom free). The validation of the French version of the SCL-90-R demonstrated satisfactory reliability and validity [58].

## Data analyses

We conducted a CFA for the three-factor model reported by Campbell et al. [15]. The goodness-of-fit indices employed were chi-square, the comparative fit index (CFI), the Tucker-Lewis index (TLI), the standardized root mean residual (SRMR) and the root mean square error of approximation (RMSEA). The values of CFI and TLI should be above 0.90 for an acceptable fit and above 0.95 for a good fit, and the SRMR and the RMSEA should be under 0.08 for an acceptable fit and under 0.05 for a good fit [59,60]. The internal consistency of the obtained subscales was estimated by calculating Cronbach's alpha and McDonald's omega coefficients, which are deemed acceptable above 0.7. Agreement, that is, the capacity of the scale to produce the same score for an individual at two different times, was tested with intraclass correlation coefficients (ICCs). ICCs above 0.74 were considered good, and ICCs between 0.60 and 0.74 were considered acceptable [61–63]. Reliability, that is, the capacity of the scale to produce the same ranking for participants at two different times, was estimated using test–retest correlations. Test–retest correlations above 0.7 were considered strong, above 0.6 good, and above 0.5 acceptable [64,65]. Agreement and reliability were calculated based on a repeat survey at 1 year after the original survey. Criterion-related validity was established by examining Spearman's correlations between the ETMCQ subscales and three related constructs used in the original study [15]: childhood traumatic experiences (CTQ-SF), mentalizing capacities (RFQ-8) and attachment style (ECR-R). Additionally, correlations of the ETMCQ subscales with loneliness (UCLA-LS) and psychopathology (SCL-90-R) were examined. The correlations were interpreted in terms of effect sizes rather than statistical significance alone given the sample size and potential inflation of correlations due to shared method variance [66,67]. Effect sizes between 0.1 and 0.3 were considered weak, between 0.3 and 0.5 moderate, and above 0.5 strong [68,69]. A non-parametric Mann-Whitney U test for two independent groups was used to evaluate differences in the ETMCQ subscales according to gender. Regarding attachment styles, individuals were categorized according to the four categories proposed by Fraley et al. [50] and replicating the method of categorization proposed by Campbell et al. [15]. After determining the median scores in the anxiety and avoidance dimensions over the whole sample, individuals were ranked as having "low" or "high" anxiety, and "low" or "high" avoidance, depending on whether their scores were below or above the median scores. Then, individuals with low anxiety and low avoidance were categorized as "Secure", those with low anxiety and high avoidance as "Dismissing", those with high anxiety and low avoidance as "Preoccupied", and those with high anxiety and high avoidance as "Fearful". One-way ANOVAs (Welch's) and Dwass-Steel-Crichlow-Fligner pairwise comparisons were conducted to assess differences in ETMCQ subscales between the four distinctive attachment styles. A generalized linear model of mediation was used to evaluate the mediation of the ETMCQ subscales between childhood traumatic experiences (CTQ-SF total score) and psychopathology (SCL-90-R GSI). Another generalized linear model of mediation evaluated the mediation of the ETMCQ subscales between loneliness (UCLA-LS) and psychopathology (SCL-90-R GSI), after considering the correlation between loneliness and psychopathology. All *p*-values were adjusted with Bonferroni corrections. All statistical analyses were performed using Jamovi Desktop 2.3.28 [70] and JASP 0.17.1.0 [71].

## Results

### Demographic data

Our sample consisted of 161 women (53.3%) and 141 men (46.7%). Mean age was 37.1 years (SD = 12.3), with women having a mean age of 37.3 years (SD = 13.6) and men 36.8 years (SD = 10.7). The mean SCL-90-R GSI was 0.71, which is in line with what is found in general populations [72–74]. The mean CTQ-SF total score was 42.5, which corresponded to previous findings in non-clinical populations [47,75]. The sociodemographic characteristics of the sample are summarized in Table 1.

**Table 1. Sociodemographic characteristics of the sample.**

|  |  | n | % |
|---|---|---|---|
| **Age range** | 18-25 | 75 | 26 |
|  | 26-35 | 75 | 26 |
|  | 36-45 | 67 | 23 |
|  | 46-55 | 52 | 18 |
|  | 56-65 | 21 | 7 |
|  | >65 | 3 | 1 |
| **Ethnicity** | Asian | 9 | 3 |
|  | Black/African/other | 11 | 4 |
|  | Mixed | 14 | 5 |
|  | Other | 7 | 2 |
|  | White | 250 | 85 |
|  | PNTS | 2 | 1 |
| **Education level** | Compulsory | 11 | 4 |
|  | Vocational | 9 | 3 |
|  | High school | 41 | 14 |
|  | Higher education | 37 | 13 |
|  | University | 195 | 65 |
| **Sexual orientation** | Heterosexual | 238 | 81 |
|  | Homosexual | 13 | 4 |
|  | Other | 35 | 12 |
|  | PNTS | 7 | 2 |
| **Marital status** | Single | 94 | 32 |
|  | Partnered | 115 | 39 |
|  | Married | 70 | 24 |
|  | Separated | 2 | 1 |
|  | Divorced | 11 | 4 |
|  | Widowed | 1 | 0.3 |
| **Work status** | Homemaker | 5 | 2 |
|  | Trainee | 41 | 14 |
|  | Part-time job | 50 | 17 |
|  | Full-time job | 163 | 56 |
|  | Unemployed | 21 | 7 |
|  | Disabled not working | 5 | 2 |
|  | Retired | 8 | 3 |

Note. PNTS = prefer not to say

## Factor structure, internal consistency and reliability

In line with Campbell et al. [15], item responses were treated as ordinal variables, and CFA used a polychoric correlations estimation method, namely diagonal weighted least squares [76]. CFA results were examined looking first at models' goodness-of-fit indices (GFIs) (Table 2), then at items parameters estimates (Table 3).

Model fit improvement was driven by item modification indices (MIs) and model GFIs [77]. Table 2 shows the results of the successive CFA models. The initial 15-item, 3-factor model did not provide an acceptable fit to the data (Table 2). To improve model fit, first we analyzed residual covariances between items in the same subscales and checked them for semantic similarity, leading us to correlate the residuals of four items. After this step, GFIs were improved but still below acceptance thresholds (Table 2). Second, we analyzed MIs. Items 1 (Trust, "I usually ask people for advice when I have a personal problem") and 3 (Mistrust, "I'd prefer to find things out for myself on the internet rather than asking people

**Table 2. Goodness-of-fit measures.**

| Model | CFI | TLI | SRMR | RMSEA |
|---|---|---|---|---|
| 3-factor, 15 items | 0.894 | 0.872 | 0.104 | 0.129 (0.118–0.139) |
| 3-factor, 15 items, 4 CRs | 0.924 | 0.904 | 0.095 | 0.111 (0.100–0.123) |
| 3-factor, 13 items, 4 CRs | 0.960 | 0.946 | 0.077 | 0.088 (0.074–0.101) |
| 3-factor, 12 items, 4 CRs | 0.978 | 0.969 | 0.062 | 0.062 (0.046–0.079) |
| Female (n = 161) | 0.975 | 0.965 | 0.076 | 0.069 (0.044–0.093) |
| Male (n = 141) | 0.970 | 0.957 | 0.077 | 0.074 (0.043–0.097) |

Note. CRs = Correlated residuals; CFI = comparative fit index; TLI = Tucker-Lewis Index; SRMR = standardized root mean square residual; RMSEA = root mean square error of approximation.

**Table 3. ETMCQ individual item psychometric characteristics with loadings according to the models tested.**

| Subscale | Item | Mean | SD | Skewness | Kurtosis | Initial model 15-item | | | Final model 12-item + 4CR | | |
|---|---|---|---|---|---|---|---|---|---|---|---|
| | | | | | | Std loading | $R^2$ | Uniq. | Std loading | $R^2$ | Uniq. |
| Trust | 1 | 3.954 | 1.615 | –0.048 | –0.977 | 0.720 | 0.518 | 0.482 | – | – | – |
| Trust | 2 | 4.911 | 1.355 | –0.732 | 0.455 | 0.701 | 0.491 | 0.509 | 0.713 | 0.508 | 0.492 |
| Trust | 7 | 4.699 | 1.491 | –0.598 | 0.01 | 0.679 | 0.461 | 0.539 | 0.638 | 0.407 | 0.593 |
| Trust | 8 | 5.222 | 1.474 | –0.776 | –0.028 | 0.509 | 0.259 | 0.741 | 0.426 | 0.181 | 0.819 |
| Trust | 13 | 4.374 | 1.709 | –0.27 | –0.749 | 0.681 | 0.464 | 0.536 | 0.675 | 0.456 | 0.544 |
| Mistrust | 3 | 5.103 | 1.542 | –0.562 | –0.425 | 0.321 | 0.103 | 0.897 | – | – | – |
| Mistrust | 4 | 4.066 | 1.629 | –0.024 | –0.837 | 0.673 | 0.453 | 0.547 | 0.640 | 0.410 | 0.590 |
| Mistrust | 9 | 4.421 | 1.646 | –0.261 | –0.801 | 0.501 | 0.251 | 0.749 | 0.483 | 0.233 | 0.767 |
| Mistrust | 10 | 3.404 | 1.739 | 0.195 | –1.032 | 0.617 | 0.381 | 0.619 | 0.587 | 0.345 | 0.655 |
| Mistrust | 14 | 3.159 | 1.468 | 0.414 | –0.289 | 0.421 | 0.177 | 0.823 | 0.362 | 0.131 | 0.869 |
| Credulity | 5 | 2.374 | 1.513 | 1.156 | 0.707 | 0.798 | 0.637 | 0.363 | 0.408 | 0.166 | 0.834 |
| Credulity | 6 | 2.699 | 1.496 | 0.604 | –0.534 | 0.694 | 0.482 | 0.518 | – | – | – |
| Credulity | 11 | 2.351 | 1.45 | 1.115 | 0.637 | 0.716 | 0.513 | 0.487 | 0.946 | 0.895 | 0.105 |
| Credulity | 12 | 2.053 | 1.39 | 1.483 | 1.636 | 0.832 | 0.692 | 0.308 | 0.622 | 0.387 | 0.613 |
| Credulity | 15 | 3.401 | 1.818 | 0.367 | –0.987 | 0.627 | 0.393 | 0.607 | 0.767 | 0.588 | 0.412 |

Note. $R^2$: explained variance, SD: Standard Deviation, Uniq.: Uniqueness. Skewness between –0.5 and +0.5 was considered not problematic, <–0.5 or>+0.5 moderately problematic; <–1.0 or>+1.0 strongly problematic; kurtosis between –2.0 and + 2.0 was considered not problematic.

for information") showed clearly the highest MIs and were removed, which improved GFIs (Table 2) but still below acceptance thresholds. Item 6 (Credulity, "When I speak to different people, I find myself easily persuaded by what they say even if this is different from what I believed before") was the item with the next highest MI; it was removed, which resulted in a 12-item model with four correlated residuals (CRs) with acceptable GFIs (Table 2). Internal consistency parameters were not uniform among the subscales. They were acceptable for ET and EC (Cronbach's α = 0.710, McDonald's ω = 0.716, and Cronbach's α = 0.744, McDonald's ω = 0.755, respectively), but below acceptance for EM (Cronbach's α = 0.590, McDonald's ω = 0.597).

All items loaded significantly ($p < 0.001$) and substantially (>0.3) on to their respective factors (Fig 1). EM and EC were significantly correlated ($r = 0.339$, $p < 0.001$), as were ET and EC ($r = 0.170$, $p < 0.05$). ET and EM were not significantly correlated (Fig 1). The mean ETMCQ subscale scores were as follows: ET, M = 4.81 (SD = 1.10); EM, M = 3.76 (SD = 1.09); EC, M = 2.54 (SD = 1.16). The Shapiro-Wilk test supported a non-normal distribution for ET ($p < 0.001$) and EC ($p < 0.01$). EM had a close to normal distribution ($p = 0.05$).

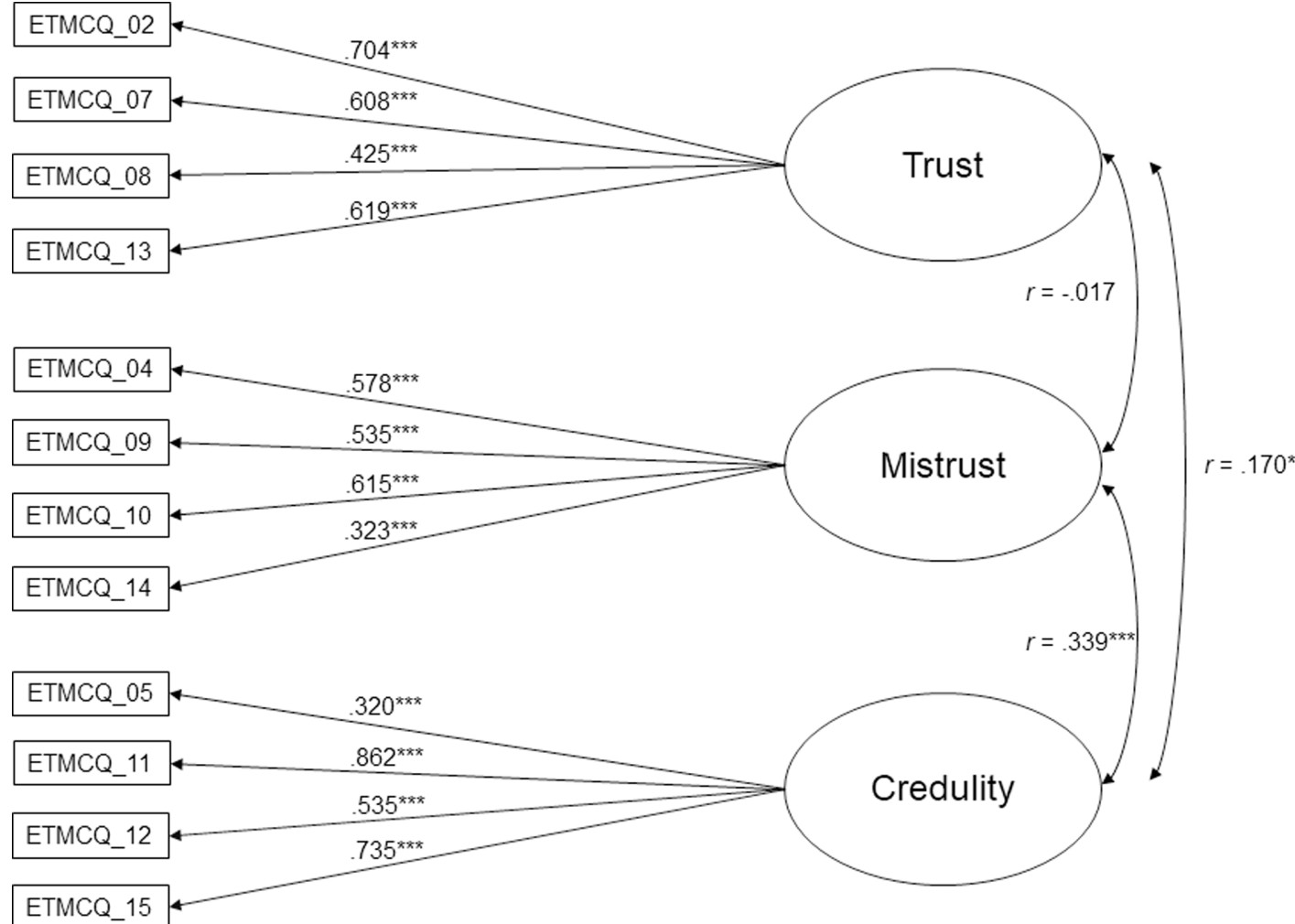

**Fig 1. CFA with factor loadings (12 items).** Note. Loadings are standardized. Rectangles indicate measured variables and circles represent latent constructs. The item numbering is retained from the original 15-item scale [15]. * $p < 0.05$. ** $p < 0.01$. *** $p < 0.001$.

A nonparametric Mann-Whitney U test for two independent groups revealed that women and men differed on both ET and EC subscales, with women showing higher scores on both subscales (ET: women M = 4.79, SD = 1.08; men M = 4.45, SD = 1.1, $p < 0.01$; EC: women M = 2.77, SD = 1.18; men M = 2.35, SD = 0.96, $p < 0.01$). EM mean scores were not different between genders. When examining Spearman correlation coefficients between the ETMCQ subscales and age, we found significant negative correlations between age and EM ($r = -0.253$, $p < 0.001$) and between age and ET ($r = -0.147$, $p <= .05$).

All 302 participants were re-contacted after a 1-year interval, and 257 took the test again (85.1% of the original sample). Between individuals who participated in the retest and those who did not, there were no significant differences in age (Mann-Whitney U = 5760.0, $p = 0.97$), gender ($\chi^2(1) = 1.69$, $p = 0.19$), SCL-90-R GSI (Mann-Whitney U = 5261.0, $p = -.33$) or ETMCQ scores (ET: Mann-Whitney U = 5079.5, $p = 0.19$; EM: Mann-Whitney U = 5459.0, $p = 0.55$; EC: Mann-Whitney U = 5144.0, $p = 0.24$). The test–retest reliability was moderate to good, with Spearman correlation coefficients of $r = 0.500$ ($p < .001$), $r = 0.600$ ($p < .001$) and $r = 0.678$ ($p < .001$) for ET, EM and EC, respectively. ICC estimates and their 95% confidence intervals were calculated based on average-rating ($k = 257$), absolute-agreement, two-way random-effects mode [78,79]. ICC results supported acceptable to strong agreement: ET = 0.706 (95% CI [0.62–0.77]); EM = 0.733 (95% CI [058–0.82]) and EC = 0.781 (95% CI [0.71–0.83]).

## Criterion-related validity

Correlations between the ETMCQ subscales and related developmental, psychological and psychopathology scales are provided in Table 4 and in S4 Appendix, and are interpreted in

Table 4. Internal consistency parameters of additional measures and Spearman's correlations with the ETMCQ subscales.

| | Cronbach's α | McDonald's ω | ET | EM | EC |
|---|---|---|---|---|---|
| **Childhood trauma (CTQ-SF)** | | | | | |
| Physical abuse | 0.853 | 0.875 | −0.019 | 0.194* | 0.136 |
| Sexual abuse | 0.940 | 0.943 | 0.007 | 0.184* | 0.144 |
| Emotional abuse | 0.880 | 0.881 | 0.001 | 0.289* | 0.305** |
| Physical negligence | 0.714 | 0.722 | −0.093 | 0.183* | 0.152 |
| Emotional negligence | 0.906 | 0.907 | −0.123 | 0.270* | 0.153 |
| Total score | 0.931 | 0.934 | −0.067 | 0.300** | 0.257* |
| **Mentalizing (RFQ-8)** | | | | | |
| RFQ_Certainty | 0.821 | 0.827 | −0.126 | −0.324** | −0.496*** |
| RFQ_Uncertainty | 0.658 | 0.737 | 0.161 | 0.280* | 0.248* |
| **Attachment (ECR-R)** | | | | | |
| Avoidant dimension | 0.876 | 0.884 | −0.342** | 0.259* | 0.104 |
| Anxious dimension | 0.873 | 0.880 | −0.001 | 0.328** | 0.430** |
| **Loneliness (UCLA-LS)** | | | | | |
| Sum score | 0.909 | 0.919 | −0.207** | 0.397** | 0.231* |
| **Psychopathology (SCL-90-R)** | | | | | |
| Global Severity Index | 0.984 | 0.985 | 0.118 | 0.396** | 0.462** |

Note. Significant ($p < 0.05$ adjusted for Bonferroni correction) effect sizes: * weak, ** moderate, *** strong. ECR-R: Experience in Close Relationships-Revied, CTQ-SF: Childhood Trauma Questionnaire Short-Form, EC: Epistemic Credulity, EM: Epistemic Mistrust, ET: Epistemic Mistrust, RFQ-8: Reflective Functioning Questionnaire 8 items, SCL-90-R: Symptoms Check List 90 items Revised, UCLA-LS: University of California Los Angeles Loneliness Scale.

terms of effect sizes rather than statistical significance alone. Here we report only moderate (between 0.3 and 0.5) and strong (above 0.5) significant correlations ($p < 0.05$ adjusted for Bonferroni correction). Regarding childhood traumatic experiences as assessed by the CTQ-SF, EC was positively correlated with emotional abuse ($r = 0.31$) and EM was positively correlated with the total CTQ-SF score ($r = 0.30$). With regard to mentalizing abilities as measured by the RFQ-8, EM ($r = -0.32$) and EC ($r = -0.50$, strong) each had a negative correlation with the RFQ_C scale. In terms of attachment dimensions assessed by the ECR-R scale, ET was negatively correlated with the avoidance dimension ($r = -0.34$), and EM ($r = 0.33$) and EC ($r = 0.43$) were both positively correlated with the anxious dimension. With respect to psychopathology as measured by the SCL-90-R, EM ($r = 0.40$) and EC ($r = 0.46$) both showed a positive correlation with the GSI. Regarding specific symptom dimensions, EM had a positive correlation with paranoid ideation ($r = 0.45$) and EC with interpersonal sensitivity ($r = 0.46$). Finally, EM had a positive correlation with loneliness as assessed by the UCLA-LS ($r = 0.40$).

## ETMCQ and attachment style

We assessed differences in the three ETMCQ subscales between the distinctive attachment styles "Secure", "Fearful", "Preoccupied" and "Dismissing" [52] (Fig 2). First, we conducted one-way ANOVAs (Welch's) were conducted, which found that the ET, EM and EC subscales differed significantly between the different attachment styles (ET: $F = 8.65$, $p < 0.001$; EM: $F = 12.19$, $p < 0.001$; EC: $F = 13.71$, $p < 0.001$). Second, Dwass-Steel-Crichlow-Fligner pairwise comparisons were conducted. For the ET subscale, the "Secure" attachment style differed significantly from "Fearful" and "Dismissing" ($p < 0.001$), "Fearful" differed significantly from "Secure" and Preoccupied" ($p < 0.001$), "Preoccupied" differed significantly from "Fearful" and "Dismissing" ($p < 0.001$), and "Dismissing" differed significantly from "Secure" and "Preoccupied" ($p < 0.001$). For the EM subscale, the "Secure" style differed significantly from "Fearful" ($p < 0.001$) and "Preoccupied ($p < 0.01$), "Fearful" differed significantly from "Secure" ($p < 0.001$), "Preoccupied" differed significantly from "Secure" ($p < 0.01$), and "Dismissing" differed significantly from "Fearful" ($p < 0.001$). For the EC subscale, "Secure" differed significantly from "Fearful" and "Preoccupied" ($p < 0.001$), "Fearful" differed significantly from "Secure" and "Dismissing" ($p < 0.001$), "Preoccupied" differed significantly from

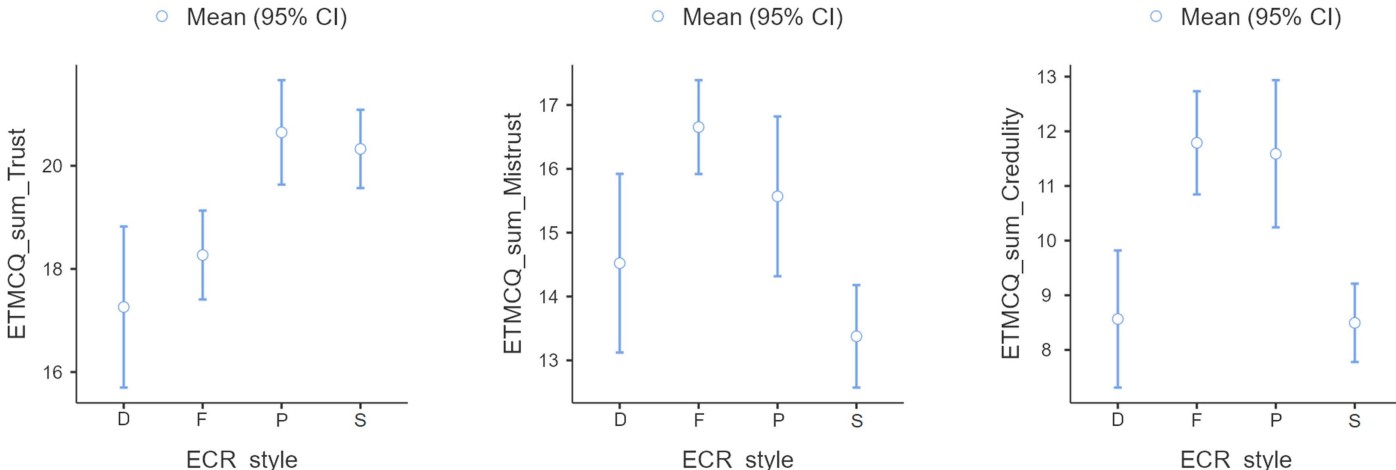

**Fig 2. Comparison between attachment styles according to the ETMCQ subscales.** Note. **D** = "Dismissive", **F** = "Fearful", **P** = "Preoccupied", **S** = " Secure".

"Secure" and "Dismissing" ($p < 0.001$), and "Dismissing" differed significantly from "Fearful" and "Preoccupied" ($p < 0.001$).

### ETMCQ and childhood traumatic experiences

A generalized linear model of mediation was constructed between the CTQ-SF total score, the ETMCQ subscales and SCL-90-R GSI (Fig 3). We found that the total effect of CTQ-SF total score on SCL-90-R GSI was significant (standardized effect sizes $\beta = 0.278$, $p < 0.001$). The direct effect of CTQ-SF total score on SCL-90-R GSI was significant ($\beta = 0.133$, $p < 0.001$) but smaller than the total effect, indicating partial mediation effects. As shown on the path diagram model of mediation (Fig 3), only EM and EC appeared to mediate the relationship between childhood traumatic experiences and psychopathology (EM standardized mediation effect between CTQ-SF total score and SCL-90-R GSI: $\beta = 0.066$, $p < 0.001$; EC standardized mediation effect between CTQ-SF total score and SCL-90-R GSI: $\beta = 0.090$, $p < 0.001$). ET had no significant mediating impact. It is of note that when examining the CTQ-SF subscales, emotional abuse and emotional negligence had the greatest direct effect on psychopathology ($\beta = 0.140$, $p < 0.005$ and $\beta = 0.121$, $p < 0.001$, respectively), with EM and EC retaining significant mediating effects.

### ETMCQ and loneliness

A second generalized linear model of mediation was constructed between loneliness, the ETMCQ subscales and psychopathology (Fig 4). We found that the total effect of the UCLA-LS sum score on SCL-90-R GSI was significant (standardized effect sizes $\beta = 0.440$, $p < 0.001$). The direct effect of UCLA-LS sum score on SCL-90-R GSI was significant ($\beta = 0.337$, $p < 0.001$) but smaller than the total effect, indicating partial mediation effects. Path diagram models of mediation by EM and EC on the relationship between loneliness and psychopathology were significant and positive, and indicated a potential role for these factors as mediators of the relationship between loneliness and psychopathology (EM standardized mediation effect between UCLA-LS sum score and SCL-90-R GSI: $\beta = 0.064$,

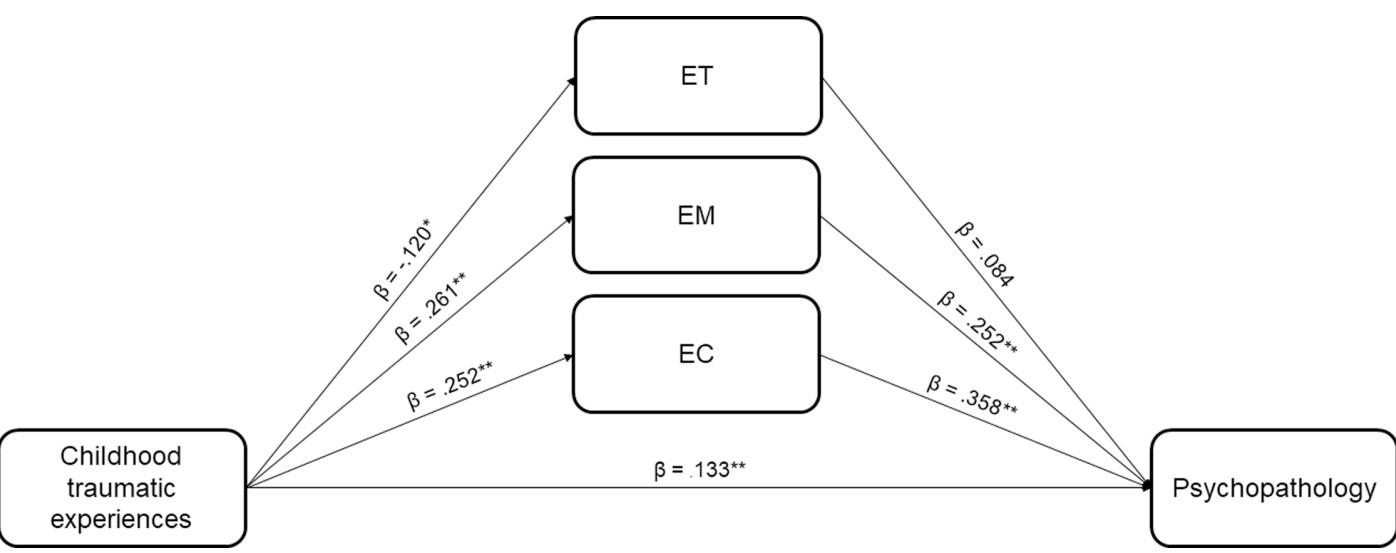

**Fig 3. Path diagram model of mediation by the ETMCQ subscales on the relationship between childhood traumatic experiences and psychopathology.** Note. $\beta$ = standardized effect size. * **p** < 0.05, ** **p** < 0.001.

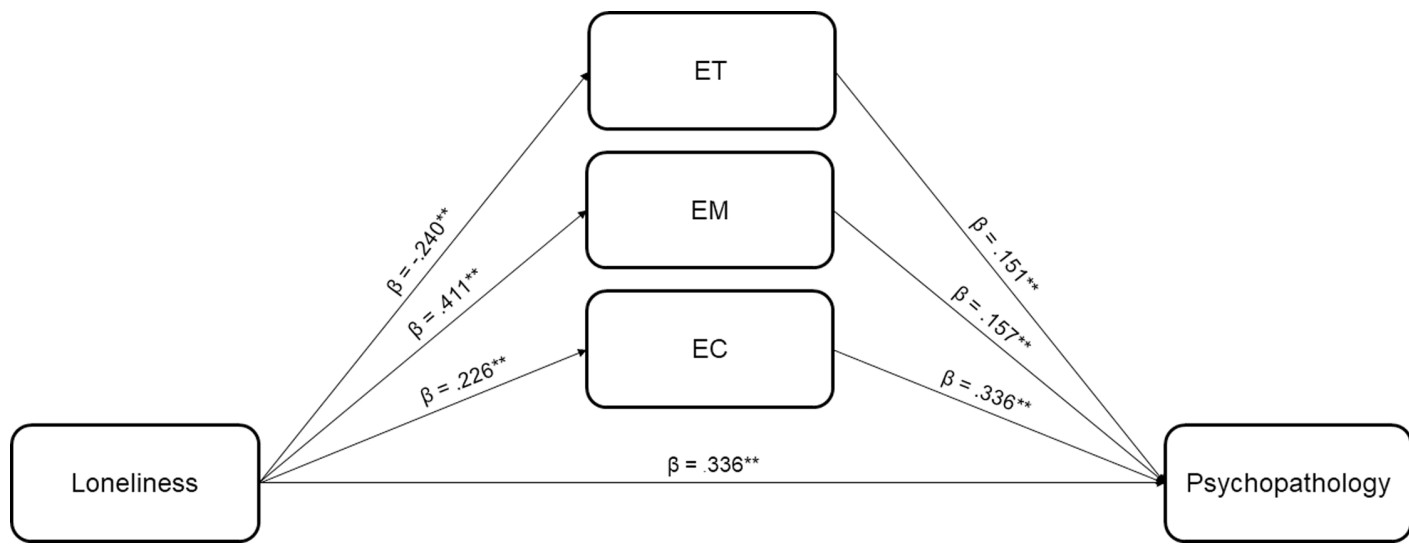

**Fig 4. Path diagram model of mediation by the ETMCQ subscales on the relationship between loneliness and psychopathology.** Note. $\beta$ = standardized effect sizes. ** $p$ < 0.001.

$p$ < 0.01; EC standardized mediation effect between UCLA-LS sum score and SCL-90-R GSI: $\beta$ = 0.076, $p$ < 0.001). ET displayed a negative significant mediation effect (ET standardized mediation effect between UCLA-LS sum score and SCL-90-R GSI: $\beta$ = −0.036, $p$ < 0.01), which indicates a likely suppressor effect, a situation in which the magnitude of the relationship between the independent variable and the dependent variable becomes larger when a mediating variable is introduced [80]. Thus, ET suppresses the relationship between loneliness and psychopathology when present, and augments it when absent. Moderation tests were also conducted, indicating a significant effect for ET ($Z$ = 2.096, $p$ < 0.05), whereas EM and EC did not moderate the association between loneliness and psychopathology.

## Discussion

The present study set out to validate the French-language version of the ETMCQ [15]. Our analyses confirmed a three-factor structure for a 12-item scale, measuring the degree of ET, EM and EC. We observed gender differences between epistemic stances, with women showing higher scores on both ET and EC subscales. Gender differences were also found in the majority of previous studies [15,16,18,19,28], although not consistently across all studies [17,20]. We observed that age had an impact on epistemic stances, with ET and EM being negatively correlated with age, as previously reported in other studies [15,17–19]. Internal consistency was acceptable for ET and EC but fell below the acceptable threshold for EM, to an extent consistent with previous studies, some of which reported satisfactory consistency for EM [15,17,22,29] whereas others, predominantly, did not [16,19,20,23,24,28,79–81]. One hypothesis for this difference is that the conceptual complexity and multidimensionality of EM compared with ET and EC (involving both maladaptive and adaptive components, cognitive and affective dimensions, and behavioural and dispositional factors) could lead to challenges in achieving consistent internal reliability [18]. Another hypothesis mentioned in the literature is the contextual variability of EM, with situational factors possibly playing a larger role in EM compared with ET and EC [19]. Individuals' levels of mistrust could fluctuate depending on context (e.g., interpersonal versus institutional trust), making it harder to capture stable

internal consistency across different samples and settings. However, we did not observe lower test–retest reliability for EM compared with ET and EC in the present study, which fails to confirm the argument that EM contains a stronger trait component. The test–retest reliability and ICCs of the subscales were acceptable to good, suggesting coherent and relatively stable individual attributes, which reflect a mostly trait-driven nature of epistemic stances measured by the ETMCQ.

Concerning criterion-related validity, the EM and EC subscales were significantly correlated with childhood traumatic experiences, insecure attachment, poor mentalizing, loneliness and psychopathology, adding weight to the hypothesis that they are both dysfunctional epistemic stances in terms of the development of mental health [15,82,83]. The most pronounced effect sizes were found between EC and the certainty dimension of mentalizing (RFQ_C) (strong), EC and global psychopathology ((SCL-90-R GSI (moderate), EM and global psychopathology (moderate), and EM and loneliness (moderate). These findings support the theoretical associations put forward in the mentalizing framework, linking dysfunctional epistemic stances, mentalizing difficulties and psychopathology, and also providing preliminary validation of the hypotheses regarding the association between epistemic stances and loneliness. Interestingly, ET was not associated with reduced levels of symptoms or with most of the predicted variables, except for avoidant attachment and loneliness. Our results add weight to converging evidence [17,20,24,28] supporting Li et al. [25], who argued that a higher level of ET is not correlated with better psychological functioning, and does not buffer against the impact of childhood adversity. Overall, we found mostly similar psychometric features to the original and the replicating ETMCQ validation studies, with possible differences in the number of items retained, mainly due to linguistic and cultural factors that need further investigation.

Our second objective was to investigate the association between epistemic trust and attachment. We found that the ETMCQ subscales were differentially associated with avoidant and anxious attachment dimensions. Our findings replicated those of Campbell et al. [15] and Asgarizadeh et al. [18], but not Liotti et al. [17] or Rodriguez et al. [19], showing that EM is more strongly correlated with the avoidance dimension and EC with the anxiety dimension. In keeping with a contemporary understanding of insecure attachment as an adaptation strategy rather than signaling maladaptation in early-life relationships, attachment avoidance can represent an attachment response to conditions that require self-reliance, whereas attachment anxiety might be better suited to unpredictable environments [18,84]. As a result, attachment avoidance may lead to EM and a shutting down of the ability to seek knowledge from others, whereas attachment anxiety tends to lead to EC by keeping that channel open, albeit at the cost of autonomy [15,18]. We also largely replicated Campbell et al.'s [15] findings concerning attachment styles. It appears that the "Dismissive" attachment style seems to protect against high levels of EM and EC, but it also was associated with the lowest level of ET, which might hinder the possibility to learn from experience when the attribution of trust is so scarce [25]. Participants classified as "Fearful" in attachment style scored both high on EM and EC and low on ET; this is the least favorable pattern of epistemic trust stances, echoing so-called epistemic petrification, in which all social-communicative channels are closed off [4,15]. Participants classified as having a "Preoccupied" attachment style scored high on EM, EC and ET, which may regularly induce conflict between the wish to trust, the experience of mistrust, and the actual attribution of trust in social interactions. Indeed, while epistemic trust allows the acceptance of information from others, epistemic vigilance wards off information from untrusted sources, and the balance between the two creates functional and/or potentially dysfunctional patterns of trust allocation that determine the readiness of the individual's epistemic channels to receive and absorb social information [4,9,22]. The contemporary epistemic

trust paradigm hypothesizes close links between attachment and epistemic trust, with early adversity impacting on the development of social-communicative abilities and conferring patterns of mistrust and/or credulity, first to attachment figures [14] and then to other significant people in the individual's social environment, preventing the individual from learning to adaptively navigate and learn from different social contexts [85,86].

Our third objective was to investigate the potential mediating role played by the epistemic stances in the relationship between childhood trauma and psychopathology. Our results regarding childhood trauma replicated those of Campbell et al. [15]. Specifically, we found that EM and EC partially mediated this link, whereas ET was not a mediator. Among the five different types of traumatic experiences measured, emotional abuse and emotional negligence had the greatest direct effect on psychopathology. Combined with our finding that EM and EC most significantly correlated with emotional abuse and negligence, our results add evidence to support the hypothesis that the emotional dimensions of child traumatization significantly disturb development at least in part through social-communicative disruptions [84,87,88]. Interestingly, the association between ET and other relevant constructs (mentalizing, mental health and personality functioning) was non-significant or very weak, as was reported in a number of studies testing the ETMCQ in non-clinical populations [16–18,24,29,89]. One possible interpretation could be that non-clinical samples are more protected against higher levels of childhood traumatization. Future studies are required to thoroughly examine this issue in clinical populations. Preliminary studies have reported that ET is increased in clinical populations, and it has been suggested that these findings may best be understood as indicating a help-seeking state in these participants [20,90]. Other hypotheses suggest either that the ET subscale does not appropriately evaluate the trust dimension or that it has a U-shaped curve (with very high and very low scores both indicating a dysfunctional epistemic stance), generating a non-linear relationship to other constructs [18,89]. Overall, our findings suggest that childhood traumatic experiences interact with attachment strategies (avoidance and anxiety dimensions) and dysfunctional epistemic stances (EM and EC) in determining vulnerability to psychopathology.

Our fourth objective was to investigate the role of epistemic trust as a mediator in the well-documented relationship between loneliness and psychopathology. We observed that the ETMCQ subscales significantly mediated the association between loneliness and psychopathology. Specifically, EM and EC were partial mediators, whereas ET had a suppressor effect, meaning that it has the potential to hinder paths between loneliness and psychopathology when it is present. These observations were corroborated by the fact that EM and EC did not moderate the association between loneliness and psychopathology. Our results may relate to other psychological models of loneliness, such as that advanced by Cacioppo et al. [34], who put forward arguments that resonate with the epistemic trust–mentalization-based framework, particularly the concept of epistemic petrification [4]. Starting from a cognitive behavioral standpoint, Cacioppo et al. [34] hypothesized a self-reinforcing loop that would sustain the formation and maintenance of chronic loneliness. They proposed that loneliness can increase hypervigilance and cognitive biases towards social threat, leading lonely individuals to anticipate negative social interactions and remember more negative social information. As a result, lonely individuals may exhibit hostile or pessimistic behaviors, which elicit exactly the unwanted responses from others that confirm their negative expectations. This loop has short-term self-protective features but, over the long term, the overall consequence is that a negative self-image is established, along with a desire to avoid social contact, resulting in chronic feelings of loneliness [35]. Future research could integrate the two models to build hypotheses on the dynamics between loneliness, trust attribution and vulnerability to psychopathology.

Our study has some limitations that should be taken into consideration. First, our sample size is relatively small in comparison to other validation studies. Although the sample size provided sufficient power for questionnaire validation, it was too small for invariance testing and results might be conflated with the potential measurement bias of the ETMCQ. Second, our sample was community-based and was predominantly White, with high levels of education; therefore, our hypotheses should be evaluated in more heterogeneous groups, including participants with clinical disorders. Previous research has suggested that epistemic trust's patterns of correlation with relevant constructs such as mentalizing, mental health and personality functioning may differ between clinical and non-clinical samples [16–18,24,29,89]. Third, our study was conducted using an online survey. Although this method is becoming more widespread in the field of psychology research, it may lead to the exclusion of part of the population, resulting in some limitations in terms of the generalizability of our study. Fourth, our findings focused only on self-report measures, which have inherent constraints, particularly the potential for correlations to be influenced and increased by response styles such as social desirability, acquiescence bias or similar reporting styles, rather than being true associations between constructs [66]. Our introduction of items controlling for social desirability in our self-report questionnaires might have only partly mitigated this limitation. Fifth, we did not examine the convergent validity of the ETMCQ with the two other existing measures of epistemic trust, the Epistemic Trust Assessment [6] and the Questionnaire Epistemic Trust [91]. Finally, our study used a cross-sectional design and thus the direction of effect for the two mediation models cannot be formally established. Therefore, as mentioned by Campbell et al. [15], future longitudinal studies should test these directions and investigate the long-term effects of epistemic trust on mental health symptoms and psychological constructs.

## Conclusions

Epistemic trust is thought to play a foundational role as a resilience factor against the development and maintenance of psychopathology through its salutogenic effect in sustaining adaptive social learning. In line with the findings of previous validation studies [15,17–19], the results of our study suggest that the ETMCQ is a valid and promising instrument that can be used to assess epistemic trust in Francophone populations. We also present preliminary evidence suggesting that epistemic trust mediates the well-established association between loneliness and psychopathology.

## Supporting information

**S1 Appendix. ETMCQ items in French.**
(PDF)

**S2 Appendix. Study dataset 1.** ETMCQ french_for CFA.
(CSV)

**S3 Appendix. Study dataset 2.** ETMCQ french_for convergence.
(CSV)

**S4 Appendix. Correlation Matrix.**
(CSV)

## Author contributions

**Conceptualization:** Christian Greiner, Paco Prada, Martin Debbané.

**Data curation:** Vincent Besch.

**Formal analysis:** Christian Greiner, Vincent Besch.

**Investigation:** Christian Greiner, Vincent Besch.

**Methodology:** Christian Greiner, Vincent Besch.

**Project administration:** Paco Prada, Martin Debbané.

**Supervision:** Paco Prada, Martin Debbané.

**Validation:** Paco Prada, Martin Debbané.

**Writing – original draft:** Christian Greiner, Vincent Besch.

**Writing – review & editing:** Marissa Bouchard-Boivin, Catherine Le Hénaff, Cécilia Von Rohr-De Pree, Nader Perroud, Paco Prada, Martin Debbané.

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
