## [Decision Letter · Decision Letter 0]

30 Jul 2024

PONE-D-24-17320Epistemic Trust, Mistrust and Credulity Questionnaire (ETMCQ) validation in French language: Investigating association with lonelinessPLOS ONE

Dear Dr. Greiner,

Thank you for submitting your manuscript to PLOS ONE. After careful consideration, we feel that it has merit but does not fully meet PLOS ONE’s publication criteria as it currently stands. Therefore, we invite you to submit a revised version of the manuscript that addresses the points raised during the review process.

We look forward to receiving your revised manuscript.

Kind regards,

Flávia L. Osório, PhD

Academic Editor

PLOS ONE

Journal Requirements:

https://journals.plos.org/plosone/article?id=10.1371%2Fjournal.pone.0251750

https://pubmed.ncbi.nlm.nih.gov/38156560/?

In your revision ensure you cite all your sources (including your own works), and quote or rephrase any duplicated text outside the methods section. Further consideration is dependent on these concerns being addressed.

Reviewers' comments:

Reviewer's Responses to Questions

**Comments to the Author**

1. Is the manuscript technically sound, and do the data support the conclusions?

Reviewer #1: Partly

Reviewer #2: Yes

2. Has the statistical analysis been performed appropriately and rigorously? 

Reviewer #1: Yes

Reviewer #2: Yes

3. Have the authors made all data underlying the findings in their manuscript fully available?

Reviewer #1: Yes

Reviewer #2: Yes

4. Is the manuscript presented in an intelligible fashion and written in standard English?

Reviewer #1: No

Reviewer #2: Yes

5. Review Comments to the Author

Reviewer #1: This manuscript provides information on the psychometric properties of the ETMCQ by examining self-report data from a sample of community-dwelling French participants. Statistical analyses focused on CFA, criterion-related validity, reliability analyses, and mediation models.

I have three major concerns about this manuscript:

- There are numerous errors regarding English grammar, sentence structure, and typos. I have specifically mentioned some of them below. Also, I may be wrong, but in some parts, I felt that the text was written in another language and then translated into English. This is a substantial issue hindering the appropriateness of this manuscript. Please have it proofread by a native English speaker. I also believe that the manuscript would significantly benefit from scientific editing.

- No rationale is provided for the decision-making process regarding improving model fit. After finding an unacceptable fit for the original CFA model, the authors immediately proceeded with item removal. I believe that item removal should be the last resort for improving fit. Strangely, they have not tried to correlate residuals first, and if and when it did not suffice, trying item removal. Additionally, on what basis did the authors remove the lowest-loading item from each subscale (i.e., why not only one item from one subscale? Why not two items from each subscale?)? They mentioned that Campbell et al. (2021) followed the same procedure, but this issue exists for their study too. In short, the statistical and theoretical rationale behind these decisions is unclear.

- The paragraphs about loneliness (in the introduction and the discussion) seem disconnected from the rest of the manuscript, which focuses on ET, attachment, and mentalizing capacity. I wonder if the mediation ideas could be explored more robustly in a separate paper rather than trying to force it to fit in with the broader measurement paper. This decision has led to a rather lengthy and sometimes incoherent manuscript. I think the authors could keep many of these variables in the paper but only utilize them as indicators of the criterion-related validity of the ETMCQ.

Minor issues:

Introduction: To my understanding, the mechanisms behind the association of ET with relevant constructs (attachment and mentalizing capacity) are not clearly delineated. For instance, how do different epistemic stances develop in early attachment relationships? How does mentalizing open the epistemic highway as an ostensive cue?

Line 33: “investigated” instead of “investigate”.

Line 37: you should put “the” before “ETMCQ” and “CTQ-SF”

Line 42: If by “external validity” you mean the association of the ETMCQ with relevant constructs, I believe the appropriate phrase would be “criterion-related validity”. Please correct this throughout the manuscript.

Lines 42-44: “We replicate previous findings that demonstrated epistemic trust’s attachment style related differences, as well as the mediating effect between childhood traumatic experiences and psychopathology.” I believe there are multiple errors in this sentence.

Line 62: “…a key sources…” Please correct the typo.

Lines 62-63: “…attributing trust to a key sources of social information, that is, to attribute epistemic trust.” I believe “trust” should be replaced with “trustworthiness” here.

Lines 64-66: Please add the page number for your direct quotation.

Lines 70-73: It may be helpful to expand on the concept of epistemic vigilance in this paragraph, as you have already mentioned it implicitly. Also, this paragraph may be too long and could be divided into two distinct paragraphs.

Lines 154-156: I wonder about the basis of your hypothesis regarding adequate internal consistency, when it is rather clear from the literature that EM does not show acceptable internal consistency coefficients (Asgarizadeh & Ghanbari, 2024).

Line 156: “We also anticipate that the ETMCQ scores converge with…” Convergent validity usually refers to the convergence between the scores of the current measure and scores of another measure aimed at the same construct.

Participants: Please describe how you determined the adequate sample size a priori. Regardless, I believe that your sample size is small relative to the current psychometric literature, which should be mentioned as a limitation of this study.

Lines 178-180: Were these bogus items or instructed-response items (see the reference below)? Also, please explain your screening in more detail (i.e., how many items were included? What was the cutoff for participant exclusion?)

Ward, M. K., & Meade, A. W. (2023). Dealing with careless responding in survey data: Prevention, identification, and recommended best practices. Annual Review of Psychology, 74(1), 577-596. https://doi.org/10.1146/annurev-psych-040422-045007

Additional measures: Please include the internal consistency coefficients of each measure in your study.

Data analyses: There is no information regarding the estimation method of the CFA. For instance, did you use ML, MLR, or WLSMV estimation methods? Moreover, you have not mentioned how you intended to examine the parameter estimates of the models (i.e., factor loadings, residual variances [uniquenesses], and R-squared values). These should be included here.

Lines 235-237: The information regarding the translation procedure is misplaced and should either be presented in the “procedures” subtitle or the “measures/materials” subtitle. Moreover, there are no paragraphs about the ETMCQ in the measures section. Either a concise paragraph should be included, or adequate information about the ETMCQ should be included in the introduction (e.g., its Likert-type scale and the number of items for each subscale).

Line 243: “…for 243 a, acceptable fit…” Please correct the typo.

Line 243: I wondered why you chose to use “internal coherence” instead of “internal consistency”.

Line 244: Please also report the Omega coefficients.

Lines 245-249: It is currently unclear as to why you have chosen to report both ICC and correlations. If both are reported to examine test-retest reliability, then there is redundancy.

Line 246: “…to control for reliability…” What does it mean to control for reliability? If I am not mistaken, “To control for” is usually used when covariates are involved.

Line 249: “Construct validity was established by examining Spearman’s correlations…” Here, you have treated the data as ordinal. As I mentioned earlier, the estimation method for the CFA models is not reported; however, it should match your decision on using the Spearman’s correlation coefficient (i.e., estimation methods that use polychoric correlations and treat data as ordinal, such as the WLSMV).

Lines 251-252: “…mentalization abilities (RFQ-8), attachment 252 style (ECR-R).” You need to put an “and” in between.

Line 253: Current practice recommendations emphasize the need to establish measurement invariance across the groups prior to conducting multigroup comparisons. In case measurement invariance is not tested, between-group differences (or lack thereof) may be products of methodological artifacts instead of actual differences. This article is an excellent review on the matter:

Putnick, D. L., & Bornstein, M. H. (2016). Measurement invariance conventions and reporting: The state of the art and future directions for psychological research. Developmental Review, 41, 71-90. https://doi.org/10.1016/j.dr.2016.06.004

Line 254: Please confirm whether invariance is being compared based on gender (i.e., how people self-identify their gender identity) or sex assigned at birth (i.e., biological differences evidenced at birth). If the authors are conflating gender with sex, please change all mentions of gender throughout the manuscript to reflect sex.

Line 258: Please expand on how you have categorized attachment styles using the ECR-R scores and cite research that previously used the same methodology.

Line 266: Please add a proper citation for the Jamovi.

Results: Correlation results are mainly discussed in terms of whether associations are "significant" or not. Reformulation of this section of the manuscript from the basis of effect size is necessary given the size of these samples and the strong likelihood that the correlations are inflated due to shared method variance. Revising the analytic plan, results, and their interpretation to employ an effect size framework with a clear delineation of benchmarks used to determine meaningful correlations in light of shared method variance would result in a stronger contribution.

Demographic data: Did you only ask participants regarding their sex/gender and age? If additional information is available (e.g., ethnicity, race, marital status, educational degree, socioeconomic status, sexual orientation, self-reported psychopathology, etc.), please report it here. If not, this is a major limitation.

Factor structure, internal consistency and reliability: Please add a table including this information: item decriptives (M, SD, skewness, and kurtosis), standardized factor loadings, residual variances (uniquenesses), and R-squared values for both the 15-item and the 12-item versions.

Line 278: “…CFI .775; TLI .728…” Equal signs should be added between the index and the value. Please correct this throughout the results.

Line 292: As you mentioned above, correlating residuals requires strong theoretical support (e.g., items measuring the same underlying construct, items having similar wordings, etc.). I believe this sort of support does not exist for correlating residuals of items 13 and 14, as they indicate different subscales.

Lines 292-293: “Internal consistency was acceptable, with Cronbach’s alpha for the entire scale at .701, and for each ETMCQ subscale as follows: Trust .710; Mistrust .690; Credulity .742.” In line 245, you proposed a cutoff of .70 for Cronbach’s alpha. Accordingly, Mistrust does not show acceptable internal consistency. The purpose of these cutoffs and benchmarks are to classify results, and adopting a “marginally adequate” stance is unacceptable. Furthermore, the problematic internal consistency of EM is reviewed and discussed in Asgarizadeh & Ghanbari (2024), and your study is not the only one finding poor values. In sum, instead of deeming the internal consistency of EM acceptable, this issue should be mentioned and expanded on in the discussion section.

Line 301: If the EM mean scores are not different across women and men, please report it explicitly rather than how it is currently implied.

Line 304: I guess by “non-parametric” you mean “non-normal”

Lines 303-305: I do not understand the relevance of the normality test here. You are reporting it between the mean differences and the correlations. Please clarify your purpose for reporting the normality of distributions, as well as their placement.

Lines 307-310: Information regarding the model parameters is misplaced and should be reported immediately after model fit indices.

Line 314: “* p < .05. ** p < .01. *** p < .001.” There are no asterisks in your figure.

Line 328: “Construct validity” is an umbrella term encompassing several types of validity. Again, I believe the appropriate phrase for association with relevant constructs would be “criterion-related validity”. Please correct this throughout the manuscript.

Lines 329-330: “Significant correlations between the ETMCQ subscales and related developmental, 330 psychological and psychopathology scales are given in Table 2.” Not all the correlations reported in Table 2 are significant.

Line 331: “…assessed in the CTQ-SF…” It should be replaced with “assessed by the CTQ-SF”.

Line 338: “…certitude subscale… incertitude subscale…” ???

Lines 329-354: I believe these paragraphs have much redundancy. If I am not mistaken, every detail you report here is presented in Table 2. Please shorten this section.

Lines 360-377: The current way of reporting pairwise comparisons is ambiguous and confusing. For instance, you have stated that “Pairwise comparisons showed significant differences between higher score for “Fearful” and “Preoccupied” styles comparing to “Secure” and “Dismissive” ones (ps < .01).” The reader would not understand the significance of each pairwise comparison. This is also not evident in Figure 2. To mitigate this, I suggest adding a table for this section which includes all of the pairwise comparisons for each of the three subscales, as well as reformulating the text.

Line 379: “Comparison between to attachment styles…” Please correct the typo.

Line 383: On line 259, you have mentioned “General Linear Model”, and here you have stated “Generalized linear model.” These two have different meanings. Please use the correct term throughout the manuscript.

Discussion: This section is usually written in the past tense.

Line 450: intra-class instead of “interclass”

Lines 469-474: With a couple of word changes, these sentences are almost fully borrowed from another source, which has not been cited here.

Lines 485-486: “…by observing that epistemic trust significantly mediates…” This confuses the reader, as your finding is exactly the opposite.

Lines 485-502: Previous articles (find below) have discussed that ET subscale may lack validity in non-clinical samples. Your findings mostly support this argument. Please expand on this matter.

Asgarizadeh, A., & Ghanbari, S. (2024). Iranian adaptation of the Epistemic Trust, Mistrust, and Credulity Questionnaire (ETMCQ): Validity, reliability, discriminant ability, and sex invariance. Brain and Behavior, 14(3), e3455. https://doi.org/10.1002/brb3.3455

Asgarizadeh, A., Hunjani, M., & Ghanbari, S. (2023). Do we really need “trust”? the incremental validity of epistemic trust over epistemic mistrust and credulity in predicting mentalizing-related constructs. Asian Journal of Psychiatry, 87, 103688. https://doi.org/10.1016/j.ajp.2023.103688

Lines 538-557: Another limitation is that you did not examine the convergent validity of the ETMCQ with other measures of ET (see below).

Schröder-Pfeifer, P., Georg, A. K., Talia, A., Volkert, J., Ditzen, B., & Taubner, S. (2022). The Epistemic Trust Assessment—An experimental measure of epistemic trust. Psychoanalytic Psychology, 39, 50-58. https://doi.org/10.1037/pap0000322

Knapen, S., Swildens, W. E., Mensink, W., Hoogendoorn, A., Hutsebaut, J., & Beekman, A. T. F. (2023). The development and psychometric evaluation of the Questionnaire Epistemic Trust (QET): A self-report assessment of epistemic trust. Clinical Psychology and Psychotherapy. https://doi.org/10.1002/cpp.2930

Reviewer #2: This manuscript provides a carefully executed replication of the study that developed the ETMCQ, enhancing the literature by integrating loneliness as a dimension of epistemic mistrust. Here are some suggestions to refine the paper further:

1. Introduction: While the introduction is thorough and scholarly, it may be perceived as too broad. Narrowing the focus to literature directly relevant to your hypotheses could streamline the reading experience. Consider relocating some of the currently cited studies to support your discussion of the results.

2. Literature Review: It is crucial to comprehensively review prior studies utilizing this instrument, especially those published in this journal. This review should address previous challenges in applying the instrument's structure and any difficulties encountered in replicating its construct validity.

3. Clarity of Hypotheses: The hypotheses should be stated more clearly. Additionally, consider using shorter paragraphs for ease of reading, though this may be a matter of personal preference.

4. Statistical Approach: The statistical methods used are suitable; however, the application of the ECR-R as a categorical measure rather than a continuous one raises questions. The section on construct validity is compelling and supports the validity of this French version of the instrument. The mediation models align with prior studies, and the novel link between loneliness, psychopathology, and mistrust is intriguing.

5. Discussion: The discussion reiterates findings extensively but falls short conceptually, especially compared to the comprehensive introduction. There seems to be minimal effort to interpret the findings within a broader psychological model. I recommend incorporating some elements from the introduction into the discussion to better explore the implications of your findings. Furthermore, the limitations section appears perfunctory. It would benefit from a more thorough exploration of the inherent constraints of self-report questionnaires, such as the potential for correlations driven by similar reporting styles rather than a genuine association between constructs.

These suggestions aim to enhance the clarity and depth of your manuscript, ensuring that it makes a significant contribution to our understanding of epistemic mistrust and its measurement.

6. PLOS authors have the option to publish the peer review history of their article (what does this mean? ). If published, this will include your full peer review and any attached files.

**Do you want your identity to be public for this peer review?** For information about this choice, including consent withdrawal, please see our Privacy Policy .

Reviewer #1: No

Reviewer #2: No

---

## [Author Response · Author response to Decision Letter 0]

22 Nov 2024

Reviewers and editor comments are responded in a "Response to Reviewers" rebuttal letter.

---

## [Decision Letter · Decision Letter 1]

27 Jan 2025

PONE-D-24-17320R1Epistemic Trust, Mistrust and Credulity Questionnaire (ETMCQ) validation in French language: exploring links to lonelinessPLOS ONE

Dear Dr. Greiner,

Thank you for submitting your manuscript to PLOS ONE. After careful consideration, we feel that it has merit but does not fully meet PLOS ONE’s publication criteria as it currently stands. Therefore, we invite you to submit a revised version of the manuscript that addresses the points raised during the review process.

We look forward to receiving your revised manuscript.

Kind regards,

Flávia L. Osório, PhD

Academic Editor

PLOS ONE

Journal Requirements:

Reviewers' comments:

Reviewer's Responses to Questions

**Comments to the Author**

1. If the authors have adequately addressed your comments raised in a previous round of review and you feel that this manuscript is now acceptable for publication, you may indicate that here to bypass the “Comments to the Author” section, enter your conflict of interest statement in the “Confidential to Editor” section, and submit your "Accept" recommendation.

Reviewer #1: (No Response)

Reviewer #2: All comments have been addressed

2. Is the manuscript technically sound, and do the data support the conclusions?

Reviewer #1: Yes

Reviewer #2: Yes

3. Has the statistical analysis been performed appropriately and rigorously? 

Reviewer #1: Yes

Reviewer #2: Yes

4. Have the authors made all data underlying the findings in their manuscript fully available?

Reviewer #1: Yes

Reviewer #2: Yes

5. Is the manuscript presented in an intelligible fashion and written in standard English?

Reviewer #1: Yes

Reviewer #2: Yes

6. Review Comments to the Author

Reviewer #1: I would like to commend the authors for their effort and dedication, which, in my view, have significantly enhanced the quality of the manuscript. The revisions have addressed several key concerns, and the paper is now much improved. I have a few minor comments and suggestions to further refine the manuscript:

1. Lines 74–75: The sentence implies that insecure attachment results in epistemic mistrust. However, as highlighted by Fonagy and colleagues, epistemic credulity could also be an outcome of insecure attachment patterns. Please revise this sentence to reflect both possibilities.

2. Line 85: The acronyms for the epistemic stances (ET, EM, and EC) are introduced here. However, in several places later in the manuscript, the full terms are used again (e.g., lines 210–213 and 482). Please ensure consistency in the use of acronyms throughout the manuscript.

3. Lines 171–173: The phrasing of your third aim is somewhat ambiguous. Please clarify it. Additionally, you mention investigating the mediating role of “epistemic trust,” but I believe you intended to refer to the three epistemic stances.

4. Lines 286–288: Performing separate CFAs for genders and attachment styles is a precursor to testing configural invariance, but it is not the test itself. Configural invariance involves testing a model with constraints imposed on factor loadings. Furthermore, scalar/weak invariance must be confirmed before conducting between-group mean comparisons. Since you have opted not to test progressive invariance models (which can be justified given the sample size), I recommend removing the section on measurement invariance from the data analysis and results. Instead, please note in the limitations section that the results might be conflated with the potential measurement bias of the ETMCQ due to the absence of invariance testing.

5. Line 340: The manuscript uses the acronym “GFIs” to refer to “Goodness of Fit Indices,” but in some instances, the singular “GFI” is used. This could lead to confusion, as “GFI” in the literature typically refers to a specific fit index. Please standardize the use of this term.

6. Factor Structure, Internal Consistency, and Reliability: The information in this section appears to be presented in the wrong order. Findings regarding model fit indices should be reported first, followed by parameter estimates (e.g., standardized loadings, R-squared values). Rearranging this section would enhance the logical flow and clarity.

Reviewer #2: The only shottcoming not picked up is the absence of a reported correlation matrix of all the vriables used in the study which could be incliuded in supplimentary materal. It would help interpretation of findings

7. PLOS authors have the option to publish the peer review history of their article (what does this mean? ). If published, this will include your full peer review and any attached files.

**Do you want your identity to be public for this peer review?** For information about this choice, including consent withdrawal, please see our Privacy Policy .

Reviewer #1: No

Reviewer #2: No

---

## [Author Response · Author response to Decision Letter 1]

29 Jan 2025

We responded to the reviewers in the "Responses to reviewers" file.

---

## [Decision Letter · Decision Letter 2]

13 Feb 2025

Epistemic Trust, Mistrust and Credulity Questionnaire (ETMCQ) validation in French language: exploring links to loneliness

PONE-D-24-17320R2

Dear Dr. Greiner

We’re pleased to inform you that your manuscript has been judged scientifically suitable for publication and will be formally accepted for publication once it meets all outstanding technical requirements.

Kind regards,

Flávia L. Osório, PhD

Academic Editor

PLOS ONE

Additional Editor Comments (optional):

Reviewers' comments:

Reviewer's Responses to Questions

**Comments to the Author**

1. If the authors have adequately addressed your comments raised in a previous round of review and you feel that this manuscript is now acceptable for publication, you may indicate that here to bypass the “Comments to the Author” section, enter your conflict of interest statement in the “Confidential to Editor” section, and submit your "Accept" recommendation.

Reviewer #1: All comments have been addressed

2. Is the manuscript technically sound, and do the data support the conclusions?

Reviewer #1: Yes

3. Has the statistical analysis been performed appropriately and rigorously? 

Reviewer #1: Yes

4. Have the authors made all data underlying the findings in their manuscript fully available?

Reviewer #1: Yes

5. Is the manuscript presented in an intelligible fashion and written in standard English?

Reviewer #1: Yes

6. Review Comments to the Author

Reviewer #1: (No Response)

7. PLOS authors have the option to publish the peer review history of their article (what does this mean? ). If published, this will include your full peer review and any attached files.

**Do you want your identity to be public for this peer review?** For information about this choice, including consent withdrawal, please see our Privacy Policy .

Reviewer #1: No

---

## [Editor Report · Acceptance letter]

PONE-D-24-17320R2

PLOS ONE

Dear Dr. Greiner,

I'm pleased to inform you that your manuscript has been deemed suitable for publication in PLOS ONE. Congratulations! Your manuscript is now being handed over to our production team.

Kind regards,

on behalf of

Dr. Flávia L. Osório

Academic Editor

PLOS ONE